# Different Actions of Intracellular Zinc Transporters ZIP7 and ZIP13 Are Essential for Dermal Development

**DOI:** 10.3390/ijms20163941

**Published:** 2019-08-13

**Authors:** Mi-Gi Lee, Bum-Ho Bin

**Affiliations:** 1Bio-Center, Gyeonggido Business and Science Accelerator, Suwon 16229, Korea; 2Department of Molecular Science and Technology, Ajou University, Suwon 443-380, Korea

**Keywords:** Zinc, ZIP7, ZIP13, connective tissue, mesenchymal stem cells, browning

## Abstract

Two mesenchymal zinc transporters, ZIP7 and ZIP13, play critical roles in dermal development. ZIP7 and ZIP13 are the closest among the conserved mammalian zinc transporters. However, whether their functions are complementary remains a controversial issue. In the present study, we found that the expression of ZIP13, but not ZIP7, is elevated by transforming growth factor beta (TGF-β) treatment, indicating that TGF-β-mediated ZIP13 amplification is crucial for collagen production during dermal development. Genome-wide gene expression analysis revealed that ~26% of genes are dependent on either ZIP7 or ZIP13, which is greater than the ~17% of genes dependent on both of them. ZIP7 depletion induces endoplasmic reticulum (ER) stress in mesenchymal stem cells, resulting in significant inhibition of fibrogenic differentiation. However, ZIP13 depletion does not induce ER stress. Though both ZIP7 and ZIP13 contain traditional ER signal peptides for their intracellular localization, their distributions are distinct. When ZIP7 and ZIP13 are coexpressed, their localizations are distinct; ZIP7 is located on the ER, but ZIP13 is located on both the ER and Golgi, indicating that only ZIP13 is a zinc gatekeeper on the Golgi. Our data illustrate that the different actions of ZIP7 and ZIP13 are crucial for dermal development.

## 1. Introduction

Zinc is an essential trace mineral in the skin, and many clinical reports have demonstrated that the skin is the first area that manifests zinc deficiency [1,2,3,4]. Because zinc is an important cofactor and structural element in numerous proteins and almost 10% of proteins in mammals show zinc association, zinc undoubtedly plays an important role in skin formation [5]. Acrodermatitis enteropathica (AE) is a rare autosomal recessive disorder and an inherited form of zinc deficiency characterized by dry and blistered skin; it is a fatal disease, and patients without proper treatment, such as zinc supplementation, die within a few years [6,7,8]. Recent advances have revealed the additional roles of zinc in living organisms, such as its activity as an intracellular second messenger, its modulation of phosphatase activity in the Fcε receptor I pathway, and its role in mammalian growth factor signaling [9]. Several stimuli induce zinc release from zinc stores within cells, which leads to intracellular zinc signaling [9,10]. Therefore, cellular zinc homeostasis is tightly regulated by proteins that transport zinc ions across lipid bilayers and the metal scavenger metallothionein (MT), which buffers excess zinc in the cytoplasm and nucleus [11,12]. Zinc deficiency and mutation of the *MSC2* gene, which encodes a metal ion transporter in secretory compartments, lead to endoplasmic reticulum (ER) dysfunction and unfolded protein response (UPR) in *Saccharomyces cerevisiae* [13]. Disruption of the *Catsup* gene, which encodes a putative secretory zinc transporter in *Drosophila*, results in protein trafficking abnormalities in Notch signaling and simultaneously induces ER stress [14]. Catsup belongs to the ZIP family, which contains 14 identified mammalian zinc transporters. Furthermore, Catsup shares high amino acid sequence similarity with the mammalian secretory zinc transporters ZIP7 and ZIP13 [11,14,15].

ZIP7 and ZIP13, the critical zinc transporters during dermis development [16,17], are classified in the LIV-1 subfamily, whose members are present in only eukaryotic cells and possess the HEXXH motif, a putative zinc-binding motif [12]. Though both ZIP7 and ZIP13 transport zinc and are involved in zinc homeostasis as intracellular zinc transporters [16,17], their precise cellular locations remain controversial. ZIP7 acts as a gatekeeper for intracellular zinc release from the ER in breast cancer cell lines [10]. ZIP7 has also been shown to regulate zinc homeostasis in the Golgi of human lung fibroblasts (WI-38 cells), prostate epithelial cells (RWPE1), erythroleukemia cells (K-562), and mammary gland epithelial cells (MCF-7) [18]. We previously reported that endogenous ZIP13 and recombinant ZIP13 are located in the Golgi in mouse fibroblasts, 293T cells, and HeLa cells [15,17,19]. Other groups have proposed that ZIP13 is located in a putative vesicular zinc store in the early secretory pathway, rather than in the ER or Golgi, and thus the loss of ZIP13 function leads to the accumulation of zinc in the vesicular zinc store, thereby reducing the zinc level in the cytoplasm and ER, resulting in ER dysfunction and stress [20]. Our recent work has shown that the loss of ZIP13 function due to pathogenic mutations does not induce ER stress [19]. These pathogenic mutations induce the rapid degradation of functional ZIP13 through the VCP/HSP90/proteasome-mediated ER-associated degradation pathway (ERAD), leading to spondylocheirodysplastic Ehlers–Danlos syndrome (SCD-EDS) due to impaired zinc homeostasis in the Golgi [19]. Surprisingly, *Drosophila* ZIP13 functions in the delivery of iron to the secretory compartments [21]. Altogether, these results indicate that despite the importance of ZIP7 and ZIP13 in secretory zinc homeostasis, whether these proteins function redundantly remains a controversial issue in zinc biology.

Here, we present evidence of the functional differences between ZIP7 and ZIP13. Our data show that secretory zinc homeostasis is tightly regulated by the different actions of the intracellular zinc transporters ZIP7 and ZIP13, both of which are indispensable for skin development and homeostasis. In addition, we found that the transforming growth factor beta (TGF-β)–SMAD–ZIP13 axis is crucial for dermal formation.

## 2. Results

### 2.1. TGF-β Induces ZIP13 Expression

The dermises of *ZIP7*-cKO and *ZIP13*-null mice were thin and exhibited reduced collagen compared to those of wild-type (WT) mice [16,19]. Because TGF-β signaling is a major factor in collagen production during dermal development, we analyzed the mRNA expression level of the well-known direct TGF-β response-related gene *SMAD7* from isolated mouse embryo cDNA. The mRNA expression level of *SMAD7* fluctuated during embryo development; its expression was high at E7 and E14 but low at E11 and E17, which is similar to the expression levels of *ZIP7* and *ZIP13* but not *Col1a2* (Figure 1A), implying that zinc transporters are associated with TGF-β signaling. We found that treatment of human mesenchymal stem cells (hMSCs) with TGF-β induced mRNA expression of *ZIP13* as *MSX2* and *SMAD7*, which are well-known TGF-β–SMAD target genes (Figure 1B) [17]. The mRNA expression of *ZIP7* was independent of TGF-β treatment, which implies that TGF-β-induced ZIP13 can support TGF-β signaling.

### 2.2. Genome-Wide Analysis Reveals Distinct Roles of ZIP7 and ZIP13

To investigate the functional similarities and distinctions between ZIP7 and ZIP13 during dermal development, we performed gene expression microarray analysis by using mRNA isolated from hMSCs after they were treated with siRNA targeting either *ZIP7* or *ZIP13* for four days. Both ZIP7 and ZIP13 protein levels were successfully reduced after siRNA-mediated knockdown (KD) [16] (Figure 2A). Interestingly, ZIP7-KD produced more dramatic changes in gene expression in hMSCs than ZIP13-KD, suggesting the predominant role of ZIP7 in hMSCs (Figure 2B). Some of the identified genes whose expression changed dramatically were validated by quantitative real-time PCR (RT-qPCR) (Figure 2C); these genes belonged to clusters 2 and 4 (Figure 2E). Gene expression analysis showed 1017 genes in ZIP7-KD hMSCs and 190 genes in ZIP13-KD hMSCs to be differentially expressed genes (DEGs) (Figure 2D). These two sets of DEGs overlapped significantly (*p*-value < 2.2 × 10^−16^ computed by Fisher’s exact test). Interestingly, most of the overlapping DEGs were downregulated in both ZIP7-KD and ZIP13-KD hMSCs, suggesting a functional commonality of ZIP7 and ZIP13 through the upregulation of common genes (cluster 7 in Figure 2E). However, a large portion of DEGs were still uniquely affected by either ZIP7- or ZIP13-KD (Figure 2D–F).

Most ZIP and ZNT family members were expressed at comparable levels in KD hMSCs and WT hMSCs (Figure 2G,H). Only ZIP14, which was recently shown to be an ER stress-inducible gene, was upregulated in ZIP7-KD hMSCs but not ZIP13-KD hMSCs (Figure 2G,H) [22].

### 2.3. Functional Analysis of ZIP7 and ZIP13

To gain insight into the functions of ZIP7 and ZIP13, we performed gene set enrichment analysis of the identified DEGs between ZIP7-KD and ZIP13-KD hMSCs using DAVID V6.8 [23]. We focused on the genes in clusters 2 and 4 (uniquely upregulated in ZIP7-KD and ZIP13-KD hMSCs, respectively) and clusters 5 and 6 (uniquely downregulated in ZIP7-KD and ZIP13-KD hMSCs, respectively) produced by comparing the DEGs in ZIP7-KD and ZIP13-KD hMSCs (Figure 2D,E) to explore the functional uniqueness of ZIP7 and ZIP13. We also focused on the genes in cluster 7, which are downregulated by both ZIP7-KD and ZIP13-KD, to explore the functional commonality of the two factors. Among the genes upregulated by ZIP7- and ZIP13-KD, those in cluster 2 were mainly enriched in processes related to the ER stress response, such as the ER unfolded protein response, the ER-associated ubiquitin-dependent process, the apoptotic process, and the response to ER, while genes in cluster 4 were mainly enriched in cysteine-type endopeptidase, blood coagulation, and wnt signaling pathway processes (Figure 3A). Among the genes downregulated by ZIP7- and ZIP13-KD, those in cluster 5 were mainly enriched in proliferation-related processes, such as cell division, DNA replication, mitotic cytokinesis, and cell proliferation, while those in cluster 6 were mainly enriched in the response to glucocorticoids and hypoxia and mRNA splicing processes (Figure 3B). The genes downregulated by both ZIP7-KD and ZIP13-KD (cluster 7) were mostly involved in inflammation and the immune response, suggesting that both are crucial in the immune network of hMSCs in a zinc-dependent and/or zinc-independent manner (Figure 3C). Taken together, these results further support the functional uniqueness and commonality of ZIP7 and ZIP13.

### 2.4. ZIP7 Is Indispensable for the Maintenance of Mesenchymal Stem Cells

To demonstrate the functional uniqueness of ZIP7 and ZIP13, we focused on their functional relationship with the ER, which was the top process determined by gene set enrichment analysis (Figure 3). A previous report demonstrated the crucial involvement of ZIP7 in the ER stress response in hMSCs [16]. In ZIP7-KD hMSCs, genes related to the UPR were upregulated by elevated ER stress. In hMSCs, the mRNAs of both *ZIP7* and *ZIP13* were readily detected. However, whether ZIP13 depletion, like ZIP7 depletion, can induce ER stress remains an unanswered question. After the treatment of hMSCs with siRNAs targeting *ZIP7* or *ZIP13* for four days, the expression levels of UPR-related genes were analyzed by quantitative real-time PCR. *ZIP7* and *ZIP13* were successfully downregulated after treatment with the corresponding siRNA (Figure 4A), indicating that the siRNAs were properly targeted to the mRNAs. The well-known UPR genes binding immunoglobulin protein (*BIP*) and CCAAT/enhancer-binding protein homologous protein (*CHOP*) were significantly upregulated in only ZIP7-KD hMSCs (Figure 4A). In addition, the ER stress-inducible mRNA encoding the zinc transporter ZIP14 was elevated in only ZIP7-KD hMSCs (Figure 4A). The growth of ZIP7-KD hMSCs was blocked by induction of the UPR, but the growth of ZIP13-KD hMSCs was normal for four days (Figure 4B). Next, the differentiation of hMSCs toward fibrogenic lineage by Masson’s trichrome stain, which stains collagen in fibrogenic-differentiated cells blue, and osteogenic lineage by Alizarin red, which stains calcium deposits in osteogenic-differentiated cells red, was monitored. The results revealed that the differentiation of hMSCs toward both lineages after siZIP13 treatment was comparable to that in siControl-treated hMSCs (Figure 4C). siZIP7 treatment significantly reduced fibrogenic and osteogenic differentiation, as previously reported (Figure 4C) [16]. Together, our results indicate that ZIP7 is indispensable for the maintenance of hMSCs via protection against ER stress, but ZIP13 is nonessential for ER function.

### 2.5. Differences in Cellular Distributions of ZIP7 and ZIP13

Our data show that ZIP7, but not ZIP13, is crucial for ER function. Both transporters contain traditional ER signal peptides in the n-, h-, and c-regions with a signal peptide peptidase cleavage site (Figure 5A). Due to the lack of appropriate antibodies for double staining, the two proteins with distinct tags were coexpressed in hMSCs to elucidate differences in their cellular distribution. Our previous data showed that the signal from ZIP7 overlaps well with the signals from ER-Tracker and the ER protein BIP, but not those from Golgi-Tracker (BODIPY^®^ TR) and the trans-Golgi network (TGN) protein [16]. We found that ZIP7 and ZIP13 showed a partially distinct cellular distribution (Figure 5B). The ZIP7 signal did not merge with that from a Golgi tracker (Figure 5C). However, the ZIP13 signal merged with the signals of both ER-Tracker and Golgi tracker (Figure 5D,E), implying that ZIP7 and ZIP13 are involved in distinct mechanisms of cellular zinc homeostasis. Finally, we propose a model for the functions of ZIP7 and ZIP13 within cells. At least in mesenchymal lineages, ZIP7 is localized to the ER, where it regulates zinc homeostasis for classical ER function (Figure 6). In contrast, ZIP13 resides in the Golgi, where it regulates zinc homeostasis.

## 3. Discussion

In the present study, we provide evidence of the functional uniqueness and commonality of ZIP7 and ZIP13. Genome-wide analysis revealed that these two proteins are distinctly involved in biological pathways. We confirmed that only ZIP7, an ER zinc transporter, plays a critical role in ER functions. ZIP13, a Golgi zinc transporter, may support collagen production via the TGF-β–SMAD–ZIP13 axis [23]. ZIP13 is elevated by TGF-β signaling and supports nuclear translocation of SMAD for its activation, which may lead to collagen production. Therefore, TGF-β-mediated ZIP13 amplification might be crucial for collagen production during dermal development.

Genome-wide analysis revealed that ~26% of genes are dependent on either ZIP7 or ZIP13 but not both, implying that they play distinct roles. A large proportion of genes dependent on only ZIP7 are related to ER functions, and ER stress-response genes were markedly increased with elevated ZIP14 after ZIP7 depletion. Recent advances have shown that ZIP14 is involved in the ER stress response and that ER stress induced by zinc deficiency upregulates ZIP14 expression to alleviate ER stress by importing zinc [24,25]. Although the transport substrate of ZIP14 remains controversial and its substrate specificity might be tissue-dependent, ZIP14 is clearly also an ER stress-response gene, as confirmed by our present data (Figure 4).

We next asked how ZIP7, but not ZIP13, is specifically involved in ER stress. Differences in the ZIP7 and ZIP13 structures have been reported [12,15]. ZIP7 contains at least two His-rich domains, one in the lumen-facing N-terminus and one in the cytosolic intracellular loop 2 (Int2), but ZIP13 does not possess Int2 [26]. Int2 of ZIP13 is predicted to adopt a secondary structure, but Int2 of ZIP7 is predicted to be an unstructured flexible loop, which is a ZIP7-specific property [15]. His-rich domains and Int2 of ZIP7 might be important for ER homeostasis.

ZIP7 and ZIP13 are major intracellular zinc transporters in hMSCs [16]. These transporters are highly expressed in fibroblasts that are differentiated from mesenchymal stem cells [16], thus suggesting that ZIP7 and ZIP13 play important roles in mesenchymal lineages. *ZIP7*-cKO/Col-cre and *ZIP13*-KO mice in mesenchymal lineages showed dysgenesis [16,17]. In the present study, we showed that ectopically expressed ZIP13 protein is present in both the ER and Golgi. Membrane proteins whose final destination is the Golgi membrane are initially inserted into the ER membrane for proper folding and then transported into the Golgi. Therefore, it is not unusual to observe Golgi-resident membrane proteins in the ER, particularly in overexpression systems. Thus, ER-resident ZIP13 might not yet be properly functional, which might indicate distinct cellular roles between ZIP7 and ZIP13. In addition, we showed that though ZIP13 is normally expressed in hMSCs, ZIP7-KD inhibits mesenchymal differentiation. Proper ER function may be important in the maintenance of undifferentiated hMSCs but not Golgi function. The importance of Golgi function may increase during mesenchymal differentiation, since the Golgi is required for the proper production and secretion of extracellular connective tissue components, such as collagen.

In conclusion, we have clarified the functional differences between ZIP7 and ZIP13 and show that their different regulation of cellular zinc homeostasis is critical for dermal development and homeostasis.

## 4. Materials and Methods

### 4.1. Cell Culture and Materials

hMSCs (Lonza, Basel, Switzerland) were cultured at 37 °C in Dulbecco’s Modified Eagle Medium (DMEM) (Gibco, Carlsbad, CA, USA) containing 10% fetal bovine serum (FBS). Fibrogenic differentiation was induced by connective tissue growth factor (CTGF) treatment [16]. Osteogenic differentiation was induced by using a StemPro^®^ Osteoblast Differentiation Kit (Gibco). Masson’s trichrome stain (Sigma, St. Louis, MO, USA) and Alizarin red (ScienCell Research Laboratories, Carlsbad, CA, USA) were used as previously described [16].

### 4.2. Transfection

Cells were transfected with 100 nM ON-TARGETplus SMARTpool ZIP7 or ZIP13 siRNAs (Thermo Fisher Scientific, Waltham, MA, USA) using Lipofectamine RNAiMAX (Thermo Fisher Scientific) according to the manufacturer’s instructions, as previously described [16]. The plasmids were transfected using Lipofectamine 2000 (Thermo Fisher Scientific) following standard procedures. A total of 5 × 10^4^ cells were transfected with 5 μg of each plasmid and 100 μL of Lipofectamine in 500 μL of Opti-MEM (Thermo Fisher Scientific) for 24 h.

### 4.3. Fluorescence Microscopy

Cells were cultured on Lab-Tek chamber slides (Thermo Fisher Scientific) and then fixed with 4% paraformaldehyde in phosphate-buffered saline (PBS), permeabilized with 0.1% Triton X-100 in PBS containing 1% bovine serum albumin (BSA) for 5 min, and incubated with each antibody: anti-V5 (Thermo Fisher Scientific), anti-FLAG (Sigma), anti-BIP (Abcam, Cambridge, MA, USA), and anti-TGN (Abcam). ER-Tracker™ Red (Thermo Fisher Scientific) and the Golgi tracker CellLight^®^ Golgi-RFP (Thermo Fisher Scientific) were used according to the manufacturer’s instructions. Fluorescence was monitored with a TCS SP2 AOBS inverted spectral confocal scanning system (Leica, Wetzlar, Germany) with an oil immersion 63× objective after secondary staining with Alexa Fluor 488-conjugated F(ab’)2 fragment of goat anti-mouse IgG (Thermo Fisher Scientific) and Alexa Fluor 594-conjugated F(ab’)2 fragment of goat anti-rabbit IgG (Thermo Fisher Scientific).

### 4.4. Quantitative Real-Time PCR (RT-qPCR)

Total RNA (1–2 μg) was reverse transcribed into cDNA using ReverTra Ace (Thermo Fisher Scientific) and an oligo(dT) primer. Gene expression was analyzed using TaqMan^®^ Universal Master Mix and TaqMan^®^ Gene Expression Assays (Applied Biosystems, Foster City, CA, USA). Mouse embryo RNAs were purchased from TaKaRa. The following primers were used for gene expression analysis: BIP (Mm00517691_m1), CHOP (Mm01135937_g1), ZIP7 (Mm00433930_m1), ZIP13 (Mm01329757_m1), and ZIP14 (Mm01317439_m1). Sample expression levels were normalized to *GAPDH* expression levels according to the 2^−ΔΔCt^ method, with ΔCt = Ct of the target gene − Ct of *GAPDH*, and ΔΔCT = ΔCT of the target sample − ΔCT of the control sample.

### 4.5. Cell Growth Assays

Cell growth assays were performed as previously described [27]. Briefly, on each day of the assay, the cells were fixed with 4% paraformaldehyde, washed with PBS, and then stained with 500 μL of 0.1% crystal violet for 10 min. The stained cells were washed with PBS, dried for 5 min, and lysed with 1 mL of 10% acetic acid. The absorbance at 590 nm was measured to obtain cell growth curves.

### 4.6. Microarray Analysis

Duplicate experiments were performed to generate the gene expression profile of ZIP13-depleted hMSCs. RNA isolation, RNA reverse transcription, amplification, and hybridization were performed as previously described [28]. The gene expression profile of ZIP7-depleted hMSCs was also obtained from the National Center for Biotechnology Information Gene Expression Omnibus database (GSE83097) and integrated with the expression profile of ZIP13-depleted hMSCs generated in this study. We identified differentially expressed genes (DEGs) as previously described [29]. Briefly, we first normalized the intensity values (log2 scale) across all samples using quantile normalization, and then used these values to select expressed genes using Gaussian mixture modeling [30]. Genes with intensities larger than 4.8835 in at least one sample were selected as expressed genes. Then, we compared the intensities of gene expression levels of ZIP7-KD and ZIP13-KD hMSCs with those of WT hMSCs by applying an integrative statistical method [31]. We obtained an adjusted *p*-value by combining the *p*-values obtained from a two-sample *t*-test and the log2 median difference test by random permutation between samples with a previously described method to compare gene expression levels of ZIP7-KD and ZIP13-KD hMSCs with those of WT hMSCs. Genes with adjusted *p*-value < 0.05 and log2 fold change larger than 0.4428 (95% value of randomized log2 fold change by sample permutation) were included in the gene expression dataset, which was deposited in the Gene Expression Omnibus database (GSE130154).

### 4.7. Gene Set Enrichment Analysis

Gene set enrichment analysis was performed using DAVID V6.8. Gene Ontology Biological Processes with a *p*-value < 0.05, and gene counts > 2 were used to select biological processes enriched by the genes in each cluster.

### 4.8. Statistical Analysis

Two-tailed Student’s *t*-tests were used to analyze differences between pairs of groups.

## Figures and Tables

**Figure 1 ijms-20-03941-f001:**
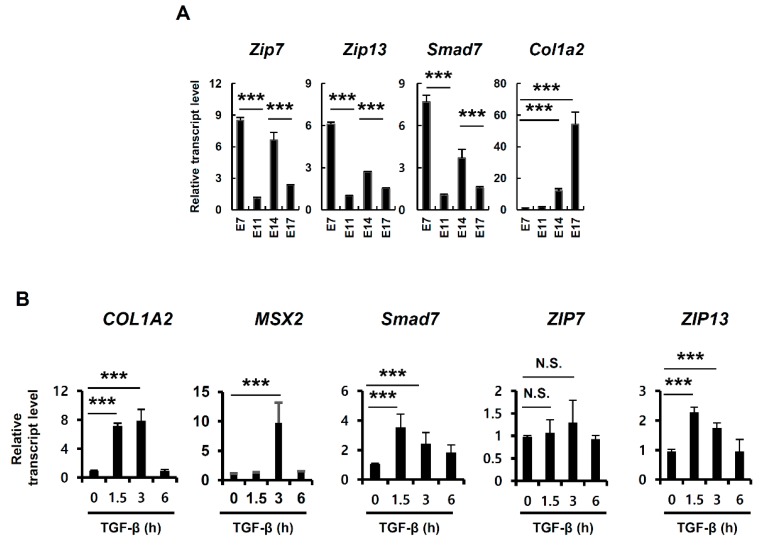
Transforming growth factor beta (TGF-β) induces *ZIP13* expression. (**A**,**B**) mRNA expression levels in mouse embryo cDNA and (**B**) human mesenchymal stem cells (hMSCs) after 20 nM TGF-β treatment. Data are representative of three independent experiments (*** *p* < 0.005).

**Figure 2 ijms-20-03941-f002:**
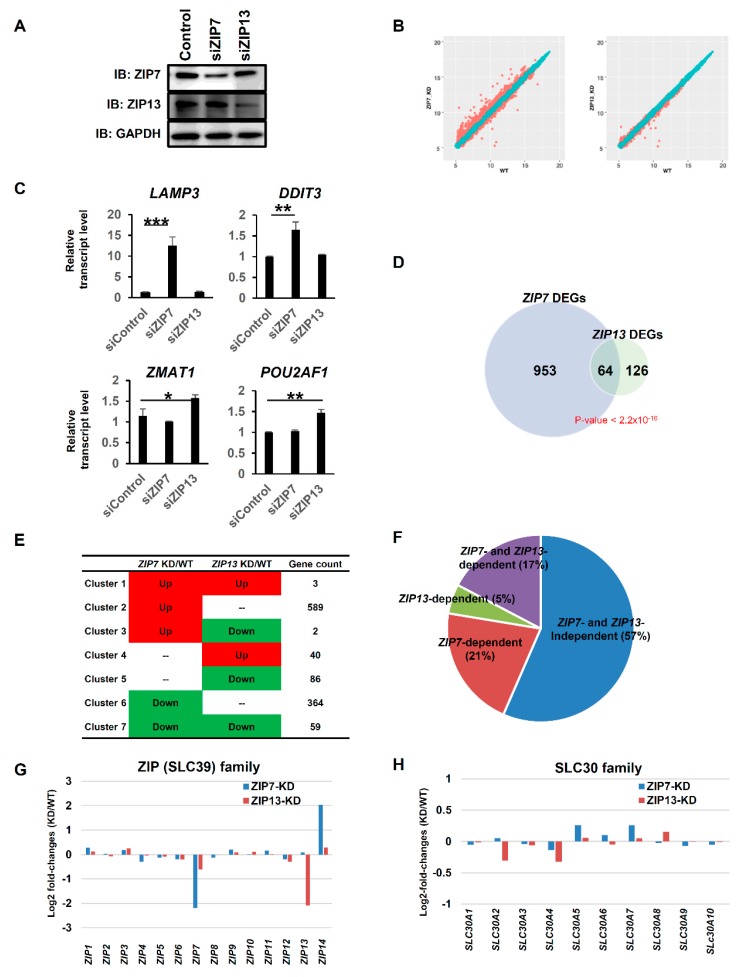
Genome-wide analysis revealed the distinct roles of ZIP7 and ZIP13. (**A**) Western blot analysis revealed that ZIP13 protein was successfully downregulated by treatment with ZIP13-targeting siRNA. (**B**) Scatter plots of averaged log2 intensities of expressed genes in wild-type (WT) versus ZIP7 knockdown (KD) and ZIP13-KD hMSCs. (**C**) Validation of identified genes whose expression was significantly changed after siRNA treatment by RT-qPCR analysis. Data are representative of three independent experiments (* *p* < 0.05; ** *p* < 0.01; *** *p* < 0.005). (**D**) Identification of differentially expressed genes (DEGs). Genes were screened for *p*-value < 0.05 and fold change > 0.4428 (95%). (**E**) Gene expression patterns determined by comparing DEGs obtained from ZIP7-KD and ZIP13-KD hMSCs. Seven clusters were obtained when the two sets of DEGs were compared. (**F**) Venn diagram based on DEGs obtained by comparing ZIP7-KD and ZIP13-KD hMSCs versus WT hMSCs. Genes with adjusted *p*-value < 0.05 and log2 fold change > 0.4428 (95% of randomized log2 fold change) are included. (**G**,**H**) Log2 fold changes in the expression of ZIP (SLC39) family members (**G**) and ZNT (SLC30) family members (**H**) are shown.

**Figure 3 ijms-20-03941-f003:**
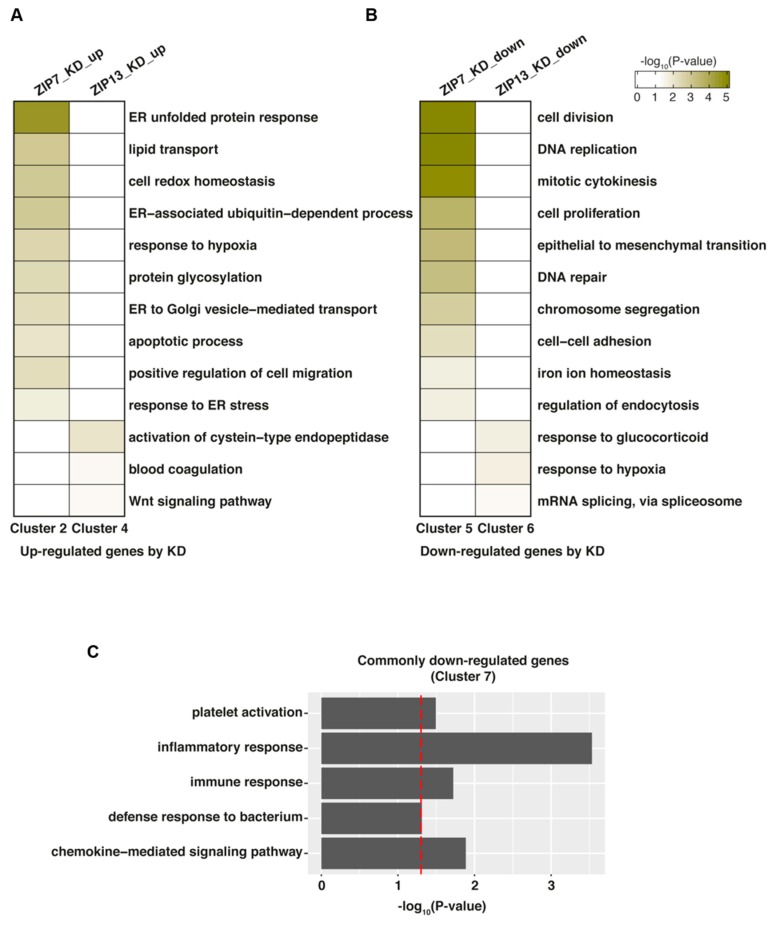
Functional analysis of ZIP7 and ZIP13. (**A**,**B**) Biological processes enriched in genes uniquely upregulated in either ZIP7-KD (cluster 2) or ZIP13-KD (cluster 4) hMSCs. (**B**) Biological processes enriched in genes uniquely downregulated in either ZIP7-KD (cluster 5) or ZIP13-KD (cluster 6) hMSCs. (**C**) Biological processes enriched in genes commonly downregulated in both ZIP7-KD and ZIP13-KD hMSCs (cluster 7). Gene Ontology Biological Process (GOBP) analysis was conducted with DAVID V6.8. Processes with a *p*-value < 0.05 and gene count > 2 were selected as representative GOBPs. The vertical red line indicates a cutoff *p*-value = 0.05.

**Figure 4 ijms-20-03941-f004:**
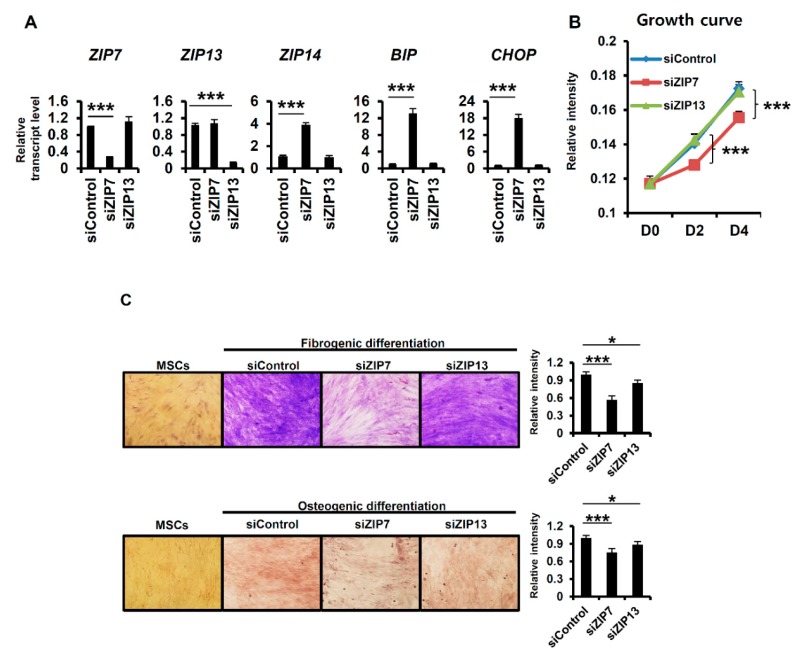
ZIP7 is indispensable for the maintenance of mesenchymal stem cells. (**A**) mRNA expression levels of zinc transporters and ER stress-response genes in hMSCs were analyzed by RT-qPCR after four days of treatment with each siRNA. Data are representative of three independent experiments (*** *p* < 0.005). (**B**) Cell growth curves after treatment with each siRNA. Data are representative of three independent experiments (*** *p* < 0.005). (**C**) Only siZIP7 disturbed the differentiation of hMSCs toward fibrogenic and osteogenic lineages. hMSCs were differentiated by their culture in differentiation medium for three weeks. Each siRNA was applied every four days. Relative intensity was calculated with ImageJ software (http://rsbweb.nih.gov/ij/download.html). Data are representative of three independent experiments (* *p* < 0.05; *** *p* < 0.005).

**Figure 5 ijms-20-03941-f005:**
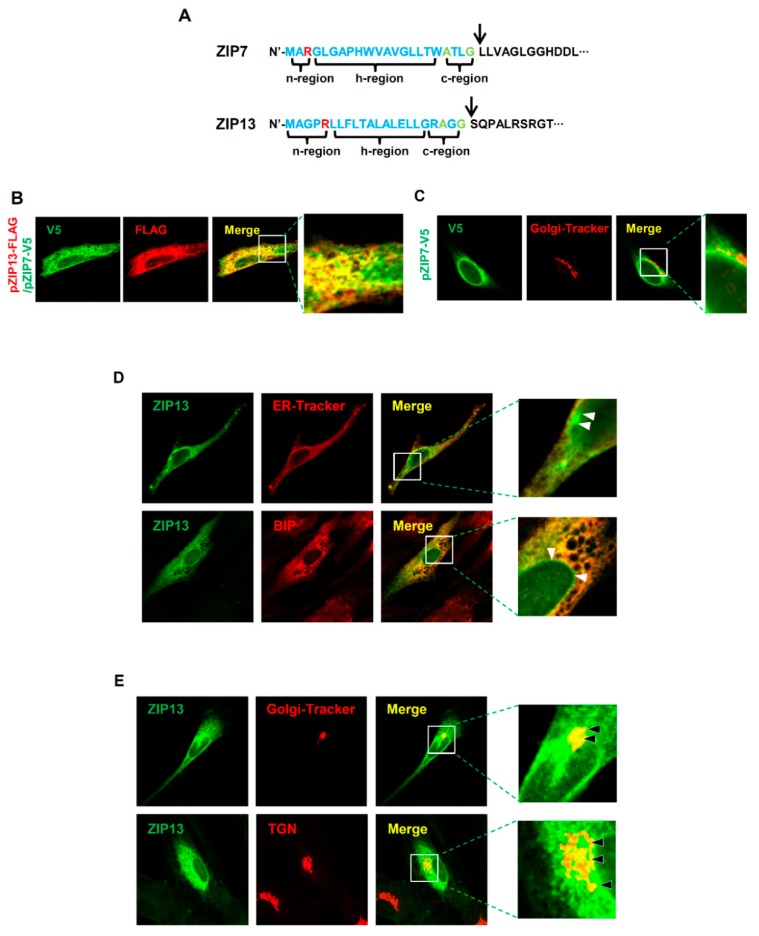
Differences in regulatory functions of ZIP7 and ZIP13. (**A**) Signal peptides of ZIP7 and ZIP13 proteins with potential signal peptidase complex cleavage sites shown by arrows. Red: positively charged amino acids; green: small amino acids conserved in the ER signal peptide. (**B**) Cellular distribution of ZIP7 and ZIP13 in hMSCs. hMSCs were transfected with plasmids encoding V5-tagged ZIP7 and FLAG-tagged ZIP13. hMSCs were stained with anti-FLAG and anti-V5 antibodies. Inset is magnified. (**C**) hMSCs were transfected with plasmid encoding V5-tagged ZIP7 and stained with anti-V5 antibody and a Golgi tracker. Inset is magnified. (**D**) hMSCs were transfected with a plasmid encoding FLAG-tagged ZIP13 and stained with anti-FLAG antibody and either an ER tracker or anti-binding immunoprotein (BIP) antibody. Inset is magnified. White arrows indicate unmerged regions. (**E**) hMSCs were transfected with a plasmid encoding FLAG-tagged ZIP13 and stained with anti-FLAG antibody and either a Golgi tracker or anti-trans-Golgi network (TGN) antibody. Inset is magnified. Black arrows indicate merged regions.

**Figure 6 ijms-20-03941-f006:**
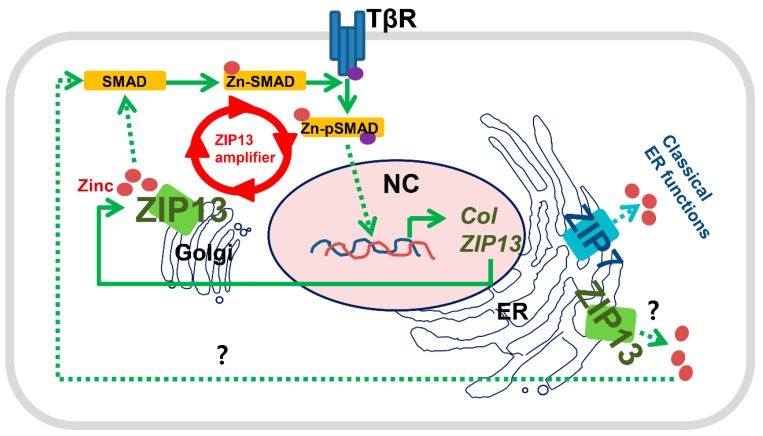
Functional differences between ZIP7 and ZIP13. ZIP7 is involved in zinc homeostasis in the ER, where it supports classical ER functions such as protein folding and modification. ZIP13 is involved in zinc homeostasis in the Golgi and associated with collagen production. ZIP13 is also expressed in the ER; however, its functional association with the ER remains unknown. Purple dots indicate phosphate involved in receptor-mediated phosphorylation.

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
