# Peer review of "Different Actions of Intracellular Zinc Transporters ZIP7 and ZIP13 Are Essential for Dermal Development"

_ijms, 2019, doi:10.3390/ijms20163941_

Round 1

Reviewer 1 Report

The paper by Lee and Bin describes experiments to ascertain the distinct role and cellular localisations of the zinc transporters Zip7 and Zip13. I find that most of the data are well described and logical, and that most of the conclusions are supported by the data. I have just a handful of minor requests of the authors at this point.

Firstly, the initial knockdowns and genome wide array analysis described in Figure 2 do not mention how many days siRNA was applied for. This needs to be comparable or consistent with the experiments in later figures. Please could the authors confirm that the duration of application in the experiments described here was for 3 or 4 days.

Secondly, in figure 3, the clusters of genes (or gene ontologies) that are compared between Zip7 and Zip13 knockdown do not mentioned any collagen synthesis pathways. I think the authors should comment on the relevant GO terms (e.g. collagen biosynthetic process). If there are no effects here in any of the comparison groups then I think an explanation as to why his might be would be merited.

Thirdly, the localisation experiments in Figure 5 are presumably performed on over-expressing cells. This is unavoidable in the absence of specific Zip antibodies but the authors should include a comment to this effect somewhere in the manuscript. They should also described better the transfection and imaging in the methods. Please describe the quantity of plasmid transfected into a defined number of cells, the number of days post-transfection before imaging, and the microscope settings employed.

Finally, for clarity in Figure 4 and on page 6 there is ambiguity about the number of days siRNA was applied for. The text (line 144) says 3 days, the figure legend (line 160) says 4 days so this should be clarified.

Additional minor suggestions:

Line 10-11 need rephrasing for English clarity

Line 20-21 should be removed as the manuscript does not indicate any translational healthcare benefits

Line 266 has a missing symbol in the qPCR experiments. My version says 1-2g of RNA was reverse transcribed.

Author Response

Reviewer #1

The paper by Lee and Bin describes experiments to ascertain the distinct role and cellular localisations of the zinc transporters Zip7 and Zip13. I find that most of the data are well described and logical, and that most of the conclusions are supported by the data. I have just a handful of minor requests of the authors at this point.

Q1: Firstly, the initial knockdowns and genome wide array analysis described in Figure 2 do not mention how many days siRNA was applied for. This needs to be comparable or consistent with the experiments in later figures. Please could the authors confirm that the duration of application in the experiments described here was for 3 or 4 days.

A: We appreciate these comments to improve our manuscript. We failed to notice this important discrepancy. We are sorry for the missing information and confusion. siRNAs were applied in the experiments for 4 days. We have amended every sentence related to this experiment in the results section, as shown below.

In section 2.2 of the Results section:

“we performed gene expression microarray analysis by using mRNA isolated from hMSCs after treatment of siRNA targeting either ZIP7 or ZIP13 for 4 days.”

In section 2.4 of the Results section:

“After the treatment of hMSCs with siRNAs targeting ZIP7 or ZIP13 for 4 days,”

Q2: Secondly, in figure 3, the clusters of genes (or gene ontologies) that are compared between Zip7 and Zip13 knockdown do not mentioned any collagen synthesis pathways. I think the authors should comment on the relevant GO terms (e.g. collagen biosynthetic process). If there are no effects here in any of the comparison groups then I think an explanation as to why his might be would be merited.

A: We appreciate these very important and critical comments. Our previous data demonstrated that despite the presence of ZIP13, ZIP7 is essential for mesenchymal stem cell survival, and its knockdown results in fewer mesenchymal stem cell-derived fibroblasts in the dermis. Based on this result, we analyzed the gene expression profiles in mesenchymal stem cells and not fibroblasts to compare the roles of ZIP7 and ZIP13. For this reason, we did not discuss the collagen synthesis pathway, which occurs in fibroblasts and not in mesenchymal stem cells. To clarify, we have added the following sentences to the Figure 6 legend.

In the Figure 6 legend:

“Functional differences between ZIP7 and ZIP13. ZIP7 is involved in zinc homeostasis in the ER, where it supports classical ER functions such as protein folding and modification mainly in mesenchymal stem cells.”

Q3: Thirdly, the localisation experiments in Figure 5 are presumably performed on over-expressing cells. This is unavoidable in the absence of specific Zip antibodies but the authors should include a comment to this effect somewhere in the manuscript. They should also described better the transfection and imaging in the methods. Please describe the quantity of plasmid transfected into a defined number of cells, the number of days post-transfection before imaging, and the microscope settings employed.

A: We appreciate these very constructive and informative comments. We completely agree with your comments. As the reviewer noted, appropriate antibodies were not available for our experiments, including double staining. Therefore, we used forced expression systems to compare the cellular distribution of ZIP7 and ZIP13. We have added this information to the Results section and increased the precision of the experimental procedures, which is crucial to understand our experiments.

In section 2.5 of the Results section:

Due to the lack of appropriate antibodies for double staining, the two proteins with distinct tags were coexpressed in hMSCs to elucidate differences in the cellular distributions of ZIP7 and ZIP13.

In the Discussion section:

“In the present study, we showed that ectopically expressed ZIP13 protein is present in both the ER and Golgi. Membrane proteins whose final destination is the Golgi membrane are initially inserted into the ER membrane for proper folding and then transported into the Golgi. Therefore, it is not unusual to observe Golgi-resident membrane proteins in the ER, particularly in overexpression systems.”

In the Materials and Methods section:

Transfection

“The plasmids were transfected using Lipofectamine 2000 (Invitrogen) following standard procedures. A total of 5 x 104 cells were transfected with 5 μg of each plasmid and 100 μL of Lipofectamine in 500 μL of Opti-MEM (Thermo Fisher Scientific) for 24 hr.”

Fluorescence microscopy section of the Materials and Methods section:

“Fluorescence was monitored with a TCS SP2 AOBS (Leica) inverted spectral confocal scanning system with an oil immersion 63× objective after secondary staining with the Alexa Fluor 488-conjugated F(ab')2 fragment of goat anti-mouse IgG (Thermo Fisher Scientific) and the Alexa Fluor 594-conjugated F(ab')2 fragment of goat anti-rabbit IgG (Thermo Fisher Scientific).”

Q4: Finally, for clarity in Figure 4 and on page 6 there is ambiguity about the number of days siRNA was applied for. The text (line 144) says 3 days, the figure legend (line 160) says 4 days so this should be clarified.

 A: We appreciate these comments. We failed to notice this important discrepancy. We are sorry for the missing information and confusion. “4 days” is correct. We have amended every sentence related to Figure 4 as “4 days”.

Additional minor suggestions:

Line 10-11 need rephrasing for English clarity

A: We thank the reviewer for this comment. We have consulted an English editing service again. If you find the English to be problematic again, please let us know, and we will enlist a different English editing service.

Line 20-21 should be removed as the manuscript does not indicate any translational healthcare benefits

A: We thank the reviewer for this comment. As the reviewer noted, lines 20-21 included overly interpreted text that has now been removed.

Line 266 has a missing symbol in the qPCR experiments. My version says 1-2g of RNA was reverse transcribed.

A: We are sorry for the confusion. We have amended this line as shown below.

RNA (1-2 g) à RNA (1-2 μg)

Again, we deeply appreciate the reviewer’s comments

Reviewer 2 Report

Zinc is an essential mineral in the development of dermis; therefore, the authors studied the roles of zinc transporters in cellular homeostasis of human mesenchymal stem cells (hMSCs). They evaluated alteration in gene expressions in ZIP7- or ZIP13-knocked down cells, and discussed coordinated actions of both transporters for dermis development, indicating TGFb-SMADs-ZIP13 axis as an important target for anti-aging strategy. Although the article suggests a physiological significance of these transporters, there is no clear evidence on the function of these Zinc transporters in hMSCs. The missing piece might be filled by showing alteration in Zing contents in hMSCs. The followings are also suggestions and comments;

1.         DEGs are shown in ZIP7- and ZIP13-KD cells in Figures 2, 3 and 4. Although mRNA expression of these transporter genes was down-regulated, the authors may have to show protein expression was also down-regulated in each cell line.

2.         ZIP13 was shown to be in almost entire cytoplasm and even at the plasma membranes (Figure 7), and co-localized with ER and Golgi trackers. ER-tracker signal was diffused in entire cells, and no clear co-localization is indicated with that; this kind of analysis should be improved. Besides, ZIP-13 was expressed in both Golgi and ER; however, Figure 7F indicates a functional expression of ZIP13 in only Golgi but not in ER. This schema is unlikely to present their results and needs to be corrected. Furthermore, ZIP7 was described to be localized in ER; however, no evidence (i.e. immunostaining result with ER tracer) is presented in the work. The authors should provide that evidence as well.

3.         The authors hypothesize the secretory transport of Zinc from ER or Golgi. Is there any experimental evidence or the rationale for that? The reviewer highly recommends measuring Zinc content in cellular spaces of hMSCs as wells as ZIP7- and ZIP13 KD cells. Also, the addition of external Zinc might be worthwhile to be tested whether these transporters are functional at the plasma membranes.  

4.         The authors insist on the importance of Zinc transporters for TGF-b signaling; however, they did not examine an effect of KD of Zinc transporters on TGF-b signaling. For example, if their hypothesis is true; does not TGF-b stimulate SMADs via TGFb receptor? Please clarify this important point because it is very easy to do, and that would be very helpful to understand the schema shown in Figure 7F.  

5.         In Figure 7F, the red small dot indicates Zinc, I guess; however, what does small purple dot stand for?

Author Response

Reviewer #2

Zinc is an essential mineral in the development of dermis; therefore, the authors studied the roles of zinc transporters in cellular homeostasis of human mesenchymal stem cells (hMSCs). They evaluated alteration in gene expressions in ZIP7- or ZIP13-knocked down cells, and discussed coordinated actions of both transporters for dermis development, indicating TGFb-SMADs-ZIP13 axis as an important target for anti-aging strategy. Although the article suggests a physiological significance of these transporters, there is no clear evidence on the function of these Zinc transporters in hMSCs. The missing piece might be filled by showing alteration in Zing contents in hMSCs. The followings are also suggestions and comments;

DEGs are shown in ZIP7- and ZIP13-KD cells in Figures 2, 3 and 4. Although mRNA expression of these transporter genes was down-regulated, the authors may have to show protein expression was also down-regulated in each cell line.

A1: We appreciate this very important comment. The protein expression of ZIP7 after siRNA-based knockdown was analyzed and reported in our previous article (J Invest Dermatol. 2017 Apr;137(4):874-883.), and we have added the results of Western blotting after the siRNA-based knockdown of ZIP13 as shown below.

In section 2 of the Results section:
“we performed gene expression microarray analysis by using mRNA isolated from hMSCs after their treatment with siRNA targeting either ZIP7 or ZIP13 for 4 days. Both ZIP7 and ZIP13 protein levels were successfully reduced after siRNA-mediated knockdown (KD) [16] (Figure 2A). ”

In the Figure 2 legend:

“Western blot analysis reveals that ZIP13 protein was successfully downregulated by treatment with ZIP13-targeting siRNA.”

ZIP13 was shown to be in almost entire cytoplasm and even at the plasma membranes (Figure 7), and co-localized with ER and Golgi trackers. ER-tracker signal was diffused in entire cells, and no clear co-localization is indicated with that; this kind of analysis should be improved. Besides, ZIP-13 was expressed in both Golgi and ER; however, Figure 7F indicates a functional expression of ZIP13 in only Golgi but not in ER. This schema is unlikely to present their results and needs to be corrected. Furthermore, ZIP7 was described to be localized in ER; however, no evidence (i.e. immunostaining result with ER tracer) is presented in the work. The authors should provide that evidence as well.

A2: We appreciate these queries. We stained the ER and Golgi with several well-known markers (J Biol Chem. 2011 Nov 18;286(46):40255-65. & J Invest Dermatol. 2017 Apr;137(4):874-883.). For example, the signal following staining of the ER with CNX (calnexin) was similar to the signal of the ER stained with BIP or ER-Tracker in our present study (J Biol Chem. 2011 Nov 18;286(46):40255-65. & J Invest Dermatol. 2017 Apr;137(4):874-883.). Though the intensity of ER staining was cellular location-dependent, the ER structure was spread throughout the entire cell, and the ER signal did not overlap with the Golgi signal at all, as shown below. In the present study, to assess our proposal with high certainty, we used two different ER markers, BIP and ER-Tracker. In addition, we verified the ER localization of ZIP7 reported in our previous study as shown below (J Invest Dermatol. 2017 Apr;137(4):874-883.). In mesenchymal stem cells, the ZIP7 signal merged with both the ER-Tracker and ER protein BIP signals. However, the ZIP7 signal did not merge with the signals from either the Golgi tracker or the Golgi protein TGN, indicating that ZIP7 is an ER protein.

From J Invest Dermatol. 2017 Apr;137(4):874-883.

We also thought that to clarify our description, we needed to add a description of our previous ZIP7 data. We added further description to the Results section as shown below.

In section 2.5 of the Results section:

“Our previous data showed that the signal from ZIP7 overlaps well with the signals from ER-Tracker and the ER protein BIP, but not those from Golgi-Tracker (BODIPY® TR) and the Golgi protein TGN [16]. We found that ZIP7 and ZIP13 showed a partially distinct cellular distribution (Figure 5B). The ZIP7 signal did not merge with that from Golgi-Tracker (Figure 5C). Instead, the ZIP13 signal merged with the signals of both ER-Tracker and a Golgi tracker (Figure 5D and E), thus implying that ZIP7 and ZIP13 are involved in distinct mechanisms of cellular zinc homeostasis.”

Reviewer #2 commented that the scheme is inappropriate. We completely agree with this. We observed localization of ZIP13 in the ER. Whether ZIP13 is functional in the ER, however, remains an unanswered question, and we thought it necessary to comment on this. Therefore, we amended the scheme to include the ER localization of ZIP13 as shown below.

In Figure 6 legend:

“ZIP13 is also expressed in the ER; however, its functional association with the ER remains unknown.”

The authors hypothesize the secretory transport of Zinc from ER or Golgi. Is there any experimental evidence or the rationale for that? The reviewer highly recommends measuring Zinc content in cellular spaces of hMSCs as wells as ZIP7- and ZIP13 KD cells. Also, the addition of external Zinc might be worthwhile to be tested whether these transporters are functional at the plasma membranes.  

A3: We appreciate this very important comment. We previously released several papers in which cellular zinc levels and zinc transport were investigated with or without either ZIP7 or ZIP13 by ICP-MS analysis and monitoring zinc-inducible genes, such as metallothionein.

PLoS Genet. 2017 Aug 30;13(8):e1006950.

J Invest Dermatol. 2017 Aug;137(8):1682-1691.

EMBO Mol Med. 2014 Aug;6(8):1028-42.

J Biol Chem. 2011 Nov 18;286(46):40255-65.

To connect with our previous study, we focused on investigating the functional differences between ZIP7 and ZIP13 in this paper. We added this information to the Introduction section as shown below.

In the Introduction section:

“Though both ZIP7 and ZIP13 transport zinc and are involved in zinc homeostasis as intracellular zinc transporters [16, 17, 19], their precise cellular locations remain controversial.”

The authors insist on the importance of Zinc transporters for TGF-b signaling; however, they did not examine an effect of KD of Zinc transporters on TGF-b signaling. For example, if their hypothesis is true; does not TGF-b stimulate SMADs via TGFb receptor? Please clarify this important point because it is very easy to do, and that would be very helpful to understand the schema shown in Figure 7F.  

A4: We appreciate this comment. Our previous data illustrated that ZIP7 depletion did not affect TGF-β signaling but induced ER stress in mesenchymal stem cells (J Invest Dermatol. 2017 Aug;137(8):1682-1691.). ZIP13 deletion was shown to disturb TGF-β signaling, and ZIP13 overexpression elevates TGF-β signaling (PLoS One. 2008;3(11):e3642.). In this study, we compared the roles of ZIP7 and ZIP13 in ER stress and in TGF-β signaling. We found that ZIP13 depletion does not induce ER stress and that TGF-β induces ZIP13 expression but not ZIP7 expression. This finding indicates that ZIP13 is not associated with ER function and that TGF-β-induced ZIP13 can support TGF-b signaling. Therefore, we suggest that the ZIP13 amplification loop is important for collagen production. We have added this information to the text as shown below.

In section 2.1 in the Results section:

“We found that the treatment of human mesenchymal stem cells (hMSCs) with TGF-β induced the mRNA expression of ZIP13 as MSX2 and SMAD7, which are well-known TGF-β-SMAD target genes (Figure 1B) [17]. The mRNA expression of ZIP7 was independent of TGF-β treatment, which implies that TGF-β-induced ZIP13 can support TGF-β signaling.

In Figure 7F, the red small dot indicates Zinc, I guess; however, what does small purple dot stand for?

A5: We are sorry for the confusion. We have added a description of the purple dots in Figure 5 legend as shown below.

In the Figure 5 legend:

“Purple dots indicate phosphate involved in receptor-mediated phosphorylation”

Reviewer 3 Report

Coordinated actions of intracellular zinc transporters ZIP7 and ZIP13 are essential for dermis development            

The manuscript aims to give a differential description of the functions and localisation of two intracellular zinc transporters, ZIP7 and ZIP13. The authors provide a series of well thought and executed experiments that provide some insight into this, no doubt, but that, in the opinion of this referee, could be improved by some complementary experiments and much more carefully redaction of the manuscript.

In particular, the gene expression microarray data are not validated by a second technique such as Q-PCR. It is not necessary to check all the genes found, but a few chosen ones would give much more weight to the reasoning afterwards.

Also, the proposed Golgi localisation of ZIP13 should be so much more convincing if some kind of pixel co-localisation analysis was provided or, alternatively, data based on co-localisation with another Golgi protein while avoiding ZIP13 overexpression conditions, so that the alledged excess in the ER would be ameliorated. As it is, a peripheral ER localisation with a possible leak to the Golgi seems a more plausible explanation for the data shown.

The authors should double-check the statistics for data on Fig. 1, panel B.

Figure 4, and its related part in materials and methods, should be corrected: In panel A, there is no indication on the normalisation method used (actin/tubulin expression?); in panel C, it is unknown  what may be the marker for fibrinogenic differentiation (collagen? alpha smooth muscle actin?) and only a look at the Mats&Meths makes us believe that the marker shown for osteogenic differentiation is alizarin red.

By far, the greatest criticism to this manuscript is the way it is writen. The first advise is to revise its english usage. Minor errors are commonly found but, worse, at times the redaction is difficult to follow. Thi is the case of the last paragraph of the discussion, where this referee understands the authors contradict themselves at several points.

The authors should tone down the conclusions. To make a more down to earth manuscript could be the major improvement this article needs. Many of the claims are loosely, and not directly, supported by the data shown. It is true that, for example, ZIP13 putative localisation in Golgi could make it a factor in lipid metabolism, or that ZIP7 dysregulation may affect dermis development,  but no data in the literature (references) or in the manuscript support this. In other words, the authors shoot too long. In this regard, the title of the manuscript is a good example of what I am refering to (the manuscript provide no data on coordination between transporters nor on their essentiallity for dermis development). The examples given are not the only ones that should be corrected, but a thorough enumeration of all the points is out of the purpose of this review. I encourage the authours to tackle this task.

Author Response

Reviewer #3

Coordinated actions of intracellular zinc transporters ZIP7 and ZIP13 are essential for dermis development            

The manuscript aims to give a differential description of the functions and localisation of two intracellular zinc transporters, ZIP7 and ZIP13. The authors provide a series of well thought and executed experiments that provide some insight into this, no doubt, but that, in the opinion of this referee, could be improved by some complementary experiments and much more carefully redaction of the manuscript.

Q1: In particular, the gene expression microarray data are not validated by a second technique such as Q-PCR. It is not necessary to check all the genes found, but a few chosen ones would give much more weight to the reasoning afterwards.

A1: We appreciate the comments on validating the microarray data. We selected several genes and performed microarray analysis and have added the new data and text shown below to the manuscript.

Figure 2:                                             

“(C) Validation of identified genes whose expression was significantly changed after siRNA treatment by RT-qPCR analysis. The data are representative of three independent experiments (*, P < 0.05; **, P < 0.01, ***, P < 0.005).”

In section 2.2 of the Results section:

“Some of the identified genes whose expression changed dramatic were validated by quantitative real-time PCR (RT-qPCR) (Figure 2C); these genes belonged to clusters 2 and 4 (Figure 2E). ”

Q2: Also, the proposed Golgi localisation of ZIP13 should be so much more convincing if some kind of pixel co-localisation analysis was provided or, alternatively, data based on co-localisation with another Golgi protein while avoiding ZIP13 overexpression conditions, so that the alledged excess in the ER would be ameliorated. As it is, a peripheral ER localisation with a possible leak to the Golgi seems a more plausible explanation for the data shown.

A2: We appreciate the reviewer’s comments, which bring up important points in our study. This question is also related to question 2 from Reviewer #1. We reanalyzed the colocalization data and added the results in Figure 5D and E. The text in the Figure 5 legend is shown below.

“White arrows indicate unmerged regions, and black arrows indicate merged regions.”

We found that ZIP13 expression was not distinct from ZIP7 expression, which was suggested by our previous data, as shown below. Furthermore, the ZIP7 signal did not overlap with the signals for the Golgi protein TGN and a Golgi tracker, as shown below. From J Invest Dermatol. 2017 Apr;137(4):874-883.

Q3: The authors should double-check the statistics for data on Fig. 1, panel B.

A3: We appreciate this comment. We analyzed and amended the statistical data.

Figure 4, and its related part in materials and methods, should be corrected: In panel A, there is no indication on the normalisation method used (actin/tubulin expression?); in panel C, it is unknown what may be the marker for fibrinogenic differentiation (collagen? alpha smooth muscle actin?) and only a look at the Mats&Meths makes us believe that the marker shown for osteogenic differentiation is alizarin red.

A1: We appreciate these comments to improve our data. We did not include these details, which are important for paper preparation. We have described the normalization method used and added information on the fibrogenic marker used to the text.

In the Materials and Methods section:

“Sample expression levels were normalized to GAPDH expression levels according to the 2–ΔΔCt method, in which ΔCt = Ct of the target gene – Ct of GAPDH, and ΔΔCT = ΔCT of the target sample – ΔCT of the control sample.

Masson’s trichrome stain (Sigma), which stains collagen in fiasson’s  differentiated cells, and alizarin red (ScienCell Research Laboratories), which stains calcium deposits in osteogenic differentiated cells, were used as previously described [16].

By far, the greatest criticism to this manuscript is the way it is writen. The first advise is to revise its english usage. Minor errors are commonly found but, worse, at times the redaction is difficult to follow. Thi is the case of the last paragraph of the discussion, where this referee understands the authors contradict themselves at several points.

A1: We appreciate these comments. We consulted an English editing service again and read the manuscript several times to find contradictory text. If language problems persist, we will receive English editing from a new company. We thank the reviewer for this comment very much.

The authors should tone down the conclusions. To make a more down to earth manuscript could be the major improvement this article needs. Many of the claims are loosely, and not directly, supported by the data shown. It is true that, for example, ZIP13 putative localisation in Golgi could make it a factor in lipid metabolism, or that ZIP7 dysregulation may affect dermis development, but no data in the literature (references) or in the manuscript support this. In other words, the authors shoot too long. In this regard, the title of the manuscript is a good example of what I am refering to (the manuscript provide no data on coordination between transporters nor on their essentiallity for dermis development). The examples given are not the only ones that should be corrected, but a thorough enumeration of all the points is out of the purpose of this review. I encourage the authours to tackle this task.

A1: We deeply appreciate this comment. We agree with the reviewer’s comments since we presented too many conclusions without proper evidence. Based on our own data, we have amended our model as shown below.

Figure 6. Functional differences between ZIP7 and ZIP13. ZIP7 is involved in zinc homeostasis in the ER, where it supports classical ER functions such as protein folding and modification. ZIP13 is involved in zinc homeostasis in the Golgi and associated with collagen production. ZIP13 is also expressed in the ER; however, its functional association with the ER remains unknown. Purple dots indicate phosphate involved in receptor-mediated phosphorylation.

We would like to again thank the reviewer for these comments!

Round 2

Reviewer 2 Report

The authors well responded to all comments and suggestions and the provided explanation is sound.

Author Response

We appreciate the comments. We checked the spells and the style. We further had English language editing service again.

Many thanks!

Reviewer 3 Report

In the present revision, the authors present a much improved manuscript. Differences are clear in both the tone and in the redaction of the text. This results in a manuscript much easier to read and understand. Also, the inclusion of controls and other suggested modifications in the figures and text is appreciated. However, unfortunately, there are a number of issues that have not been adequately responded or that have sprang out due to the modifications made. The following are a list of the major issues found.

Fig. 1, panel B. It is difficult to believe that the expression levels of the MSX2 gene at t=0 and t=3 h are statistically different after an unpaired t-test using a significance of p=0.005; the error bar for the t=3h is as big as the measured value (solid bar). The authors should re-do the experiments or use a different gene.

In page 3, line 92 the authors state: "Both ZIP7 and ZIP13 protein levels were successfully reduced after siRNA-mediated knockdown"; therefore, in Fig. 2, a WB for ZIP7 upon SiZIP7 should be included alongside that of ZIP13.

This referee appreciates the inclusion of the methods to study differentiation in the text. Nevertheless, as a general rule, a figure must include all the necessary data to understand what is being shown. Hence, the authors should state in Fig. 4C itself that they are showing/analysing collagen, for fibrinogenic differentiation, and alizarin, for osteogenic differentiation.

In Fig. 6, and on the Discussion section (page 9, line 214) the authors state that ZIP13 is at the Golgi and involved in collagen production. However, there is no support for this in Fig. 4C, where they actually show collagen production in ZIP13-devoid cells without any apparent differences compared to control. Rather, from Fig. 1B, the simplest interpretation is that ZIP13 is responsive to TGF-beta in a similar way as the genes involved in collagen production (this is, they are co-regulated), but this does not mean involvement in collagen production. To the view of this referee, ZIP13 and collagen production seem to be in two parallel branches responding to the same signalling pathway(s).

The authors keep on claiming the importance of coordination between transporters and ZIP13 for dermal development. This is observed in the title: "Coordinated actions of intracellular zinc transporters ZIP7 and ZIP13 are essential for dermal development" and again in the discussion (page 10, last paragraph starting at line 250): "In conclusion, we have clarified the functional differences between ZIP7 and ZIP13 and show that their coordinated regulation of cellular zinc homeostasis is critical for dermal development and homeostasis." In that respect, in the previous round of review, I already said that "the manuscript provide no data on coordination between transporters nor on their essentiallity for dermis development". Sadly, this has not been tackled. In particular, there is no experiment on coordination between these transporters. For example, there is no indication of a time-differential expression for those genes (e.g. first ZIP7 and later ZIP13, or the other way around) or of substitution of functions for these transporters at subsequent stages of differentiation, to name just two possibilities. Also, while the importance of ZIP13 for dermis development is clear from the literature (part of it generated by the authors), no experiments in this manuscript show a dependence of dermis development on ZIP13 functions; on the contrary, all presented evidence seems to be negative. Hence, the assertions on coordination between the two transporters and/or the importance of ZIP13 arising from this manuscript are not supported by data. The authors should modify their claims and the title of the manuscript accordingly.

Author Response

Comments and Suggestions for Authors

In the present revision, the authors present a much improved manuscript. Differences are clear in both the tone and in the redaction of the text. This results in a manuscript much easier to read and understand. Also, the inclusion of controls and other suggested modifications in the figures and text is appreciated. However, unfortunately, there are a number of issues that have not been adequately responded or that have sprang out due to the modifications made. The following are a list of the major issues found.

Q1: Fig. 1, panel B. It is difficult to believe that the expression levels of the MSX2 gene at t=0 and t=3 h are statistically different after an unpaired t-test using a significance of p=0.005; the error bar for the t=3h is as big as the measured value (solid bar). The authors should re-do the experiments or use a different gene.

A1: We appreciate this important query. We are sorry for confusing. We re-analyzed the data of the three independent experiments. The new data were inserted into Fig. 1B for MSX2.

Q2: In page 3, line 92 the authors state: "Both ZIP7 and ZIP13 protein levels were successfully reduced after siRNA-mediated knockdown"; therefore, in Fig. 2, a WB for ZIP7 upon siZIP7 should be included alongside that of ZIP13.

A2: We appreciate this comment. We performed the western blot analysis as the comments, and replaced the Fig. 2A for both expressions

Q3: This referee appreciates the inclusion of the methods to study differentiation in the text. Nevertheless, as a general rule, a figure must include all the necessary data to understand what is being shown. Hence, the authors should state in Fig. 4C itself that they are showing/analysing collagen, for fibrinogenic differentiation, and alizarin, for osteogenic differentiation.

A3: We appreciate this constructive comment that should improve our manuscript. We added the sentences  in the result section (in yellow) as below.

In the result section:

“Next, the differentiation of hMSCs toward fibrogenic lineage by Masson’s trichrome stain, which stains collagen in fibrogenic-differentiated cells blue, and osteogenic lineage by Alizarin red, which stains calcium deposits in osteogenic-differentiated cells red, was monitored. The results revealed that the differentiation of hMSCs toward both lineages after siZIP13 treatment was comparable to that in siControl-treated hMSCs (Figure 4C)."

Q4: In Fig. 6, and on the Discussion section (page 9, line 214) the authors state that ZIP13 is at the Golgi and involved in collagen production. However, there is no support for this in Fig. 4C, where they actually show collagen production in ZIP13-devoid cells without any apparent differences compared to control. Rather, from Fig. 1B, the simplest interpretation is that ZIP13 is responsive to TGF-beta in a similar way as the genes involved in collagen production (this is, they are co-regulated), but this does not mean involvement in collagen production. To the view of this referee, ZIP13 and collagen production seem to be in two parallel branches responding to the same signalling pathway(s).

A4: We thank this comment. TGF-β signaling is a well-known pathway for collagen production, and its signaling is blocked in ZIP13-/- cells by blocking the nuclear translocation of SMAD (PLoS One. 2008;3(11):e3642). Ectopic ZIP13 expression is rescued the collagen production (PLoS One. 2008;3(11):e3642). Based on these facts, we proposed that ZIP13 is involved in collagen production. However, we didn’t consider that, in present study, as the reviewer’s comment, we did not show data that support Fig 4C. So, we think that we need to tone down our proposal. Regarding the reviewer’s comment, we modified the sentences as below.

In the discussion section,

Before: ZIP13, a Golgi zinc transporter, supports collagen production via the TGF-β-SMADs-ZIP13 axis. ZIP13 is elevated by TGF-β signaling and supports nuclear translocation of SMAD for its activation, which leads to collagen production. Therefore, TGF-β-mediated ZIP13 amplification is crucial for collagen production during dermal development.

After: ZIP13, a Golgi zinc transporter, may support collagen production via the TGF-β-SMADs-ZIP13 axis [23]. ZIP13 is elevated by TGF-β signaling and supports nuclear translocation of SMAD for its activation, which may lead to collagen production.

Q5: The authors keep on claiming the importance of coordination between transporters and ZIP13 for dermal development. This is observed in the title: "Coordinated actions of intracellular zinc transporters ZIP7 and ZIP13 are essential for dermal development" and again in the discussion (page 10, last paragraph starting at line 250): "In conclusion, we have clarified the functional differences between ZIP7 and ZIP13 and show that their coordinated regulation of cellular zinc homeostasis is critical for dermal development and homeostasis." In that respect, in the previous round of review, I already said that "the manuscript provide no data on coordination between transporters nor on their essentiallity for dermis development". Sadly, this has not been tackled. In particular, there is no experiment on coordination between these transporters. For example, there is no indication of a time-differential expression for those genes (e.g. first ZIP7 and later ZIP13, or the other way around) or of substitution of functions for these transporters at subsequent stages of differentiation, to name just two possibilities. Also, while the importance of ZIP13 for dermis development is clear from the literature (part of it generated by the authors), no experiments in this manuscript show a dependence of dermis development on ZIP13 functions; on the contrary, all presented evidence seems to be negative. Hence, the assertions on coordination between the two transporters and/or the importance of ZIP13 arising from this manuscript are not supported by data. The authors should modify their claims and the title of the manuscript accordingly.

A5: We appreciate these crucial comments. We were trying to claim that both ZIP7 and ZIP13 are essential for dermal development. So, we proposed as “Coordinated~”. But, as the reviewer’s comments, we are not provided the evidences for their coordinated actions. Therefore, we thought that the title and the claiming using “Coordinated actions/regulation” have to be changed. We changed the “coordinated actions/regulation~” to the “different actions/regulation~”. We also changed the title as “Different actions of intracellular zinc transporters ZIP7 and ZIP13 are essential for dermal development.”

Round 3

Reviewer 3 Report

This reviewers acknowledges the efforts done by the authors in improving the manuscript and has no further suggestions.